# Copper Imparts a New Therapeutic Property to Resveratrol by Generating ROS to Deactivate Cell-Free Chromatin

**DOI:** 10.3390/ph18010132

**Published:** 2025-01-20

**Authors:** Salooni Khanvilkar, Indraneel Mittra

**Affiliations:** 1Translational Research Laboratory, Advanced Centre for Treatment, Research and Education in Cancer, Tata Memorial Centre, Kharghar, Navi Mumbai 410210, India; saloonikhanvilkar123@gmail.com; 2Homi Bhabha National Institute, Anushakti Nagar, Mumbai 400094, India

**Keywords:** resveratrol, copper, cell-free chromatin particles, reactive oxygen species, therapeutics, nutraceuticals

## Abstract

Resveratrol, a bioactive phytoalexin, has been extensively studied as a pharmaceutical and nutraceutical candidate for the treatment of various diseases. Although its therapeutic effects have been largely attributed to its anti-oxidant properties, its underlying mechanisms and dose dependency are not well understood. Recent studies have shown that cell-free chromatin particles (cfChPs), which are released daily from billions of dying cells, can enter circulation and be internalized by healthy cells, wherein they trigger various damaging effects, including double-strand DNA breaks. Notably, deactivating cfChPs using a mixture of resveratrol and copper can neutralize their harmful effects. The addition of copper imparts a novel therapeutic property to resveratrol viz. the generation of reactive oxygen species (ROS), which are capable of deactivating cfChPs without damaging the genomic DNA. This perspective article discusses how the deactivation of cfChPs via the ROS generated by combining resveratrol with copper can have multiple therapeutic effects. Exploiting the damaging effects of ROS to deactivate cfChPs and ameliorate disease conditions may be a viable therapeutic approach.

## 1. Introduction

Resveratrol (3,5,4′-trihydroxy-trans-stilbene) is a phytoalexin found in the fruits and leaves of several edible plants, including grapes, raspberries, blueberries, apples, and peanuts. It can also be ingested via food products such as red wine and Itadori tea [1]. In the past few decades, resveratrol has garnered much attention owing to its potential beneficial health effects on humans and its role in plant defense mechanisms [2,3]. It has been extensively studied for its pharmacological and dietary supplement applications [1,2,3,4,5]. Resveratrol has been claimed to have cardioprotective [3,6,7,8], neuroprotective [9,10,11,12], anti-inflammatory [13,14], anti-tumor [15,16,17,18,19,20,21,22], antimicrobial [23,24,25], and anti-aging [26,27] properties. Owing to its various potential health benefits, resveratrol has also gained traction as a potential therapeutic compound for treating female infertility [28] and diabetes and its related complications [29,30]. Despite the overwhelming number of studies reporting the health benefits of resveratrol, its development as an effective therapeutic compound has been limited owing to several issues such as poor bioavailability, a limited understanding of its metabolism and dose dependency, potential adverse effects, and inconsistent results across pre-clinical and clinical studies [31,32,33,34].

Studies by Fukuhara et al. [35,36] demonstrated the ability of resveratrol to cleave plasmid DNA in the presence of copper via the generation of reactive oxygen species (ROS). ROS are known to have damaging effects on proteins, lipids, and nucleic acids, including DNA [37,38,39]. In a related context, a set of studies has shown that cell-free chromatin particles (cfChPs) released from the billions of dying cells in the human body enter into circulation and can be readily internalized by healthy cells. Such internalized cfChPs can cause havoc within the host cells including genomic DNA and mitochondrial DNA and the induction of inflammatory responses and apoptotic pathways [40,41,42,43,44]. These studies also showed that the detrimental effects of cfChPs can be neutralized using nucleic acid deactivating agents, such as a combination of resveratrol and copper via the generation of ROS. Significantly, ROS could deactivate the cfChPs without causing damage to genomic DNA [45]. Collectively, these studies suggest a promising approach of exploiting the ROS generated by the combination of resveratrol and copper toward the deactivation of cfChPs to ameliorate their damaging effects.

## 2. The Damaging Effects of cfChPs

Our research group was the first to show that circulating chromatin particles can enter healthy recipient cells and integrate into and damage genomic DNA (Appendix A) [40,46,47]. Notably, cfChPs isolated from the blood of patients with cancer were found to be more active in triggering DNA damage responses and apoptosis when intravenously injected into mice compared to those isolated from healthy volunteers [40]. Furthermore, cancer cells that die during chemotherapy or radiotherapy release a large number of cfChPs, which are then readily taken up by not only the surrounding cells but also distant healthy cells that become accessible through circulation [41,48]. These cfChPs then induce DNA damage and inflammation in the healthy cells [40,41,48]. DNA damage and inflammation are the primary causes of toxicity from chemotherapy and radiotherapy and these toxicities can be ameliorated by concurrent treatment with cfChP degrading agents [42,48]. Thus, cfChPs may contribute to the harmful bystander effects observed after cancer treatment. In a more recent study, therapeutic interventions on human breast cancer xenografts were found to promote systemic dissemination of oncogenes carried via cfChPs released from dying cancer cells with the potential to induce metastasis [49]. In another study, the group found that the administration of R-Cu in an LPS-induced mouse model of sepsis prevented the release of cfChPs in the circulation and extracellular spaces of the brain, heart, and lung. R-Cu treatment prevented the release of inflammatory cytokines; DNA damage and apoptosis; dysfunction of liver and kidney; coagulopathy, fibrinolysis and thrombocytopenia; and lethality [44]. These findings implied that cfChPs released from bacterial endotoxin-induced dying host cells promote sepsis, which can be countered through the use of a cfChP deactivating agent viz. R-Cu. Considering that billions of cells die daily, the large number of cfChPs released can be imagined to act as mutagens and major contributors to degenerative conditions and aging [43,47]. Experiments have shown that the prolonged administration of a cfChP-inactivating agent can retard the biological hallmarks of aging in mice [50]. All of the above studies together indicate that the deactivation of cfChPs is an attractive therapeutic strategy to counter multiple pathologies, including cancer and aging.

## 3. Resveratrol–Copper (R-Cu) Combination as an Effective Therapeutic Solution to Deactivate cfChPs

Resveratrol and copper have individually been extensively studied for their therapeutic potential against various cancers [19,20,51,52]. The number of studies on the health benefits of resveratrol has been rapidly increasing since the late 1990s [53]. This interest is largely due to the “French paradox” wherein studies found a low rate of heart diseases among Southern French people who consume considerable amounts of red wine, despite having a high-fat diet; this was attributed to the presence of significant amounts of resveratrol in red wine (concentrations of 0.1–14.3 mg/L) [54]. Since then, several animal and pre-clinical studies have investigated the potential benefits of resveratrol in treating a myriad of health issues, including diabetes, cardiovascular diseases, cancer, and female infertility. Despite decades of research, there are several challenges to implementing resveratrol as a pharmaceutical agent. The numerous pre-clinical and clinical trials, animal studies, and meta-analyses are replete with contradictory or difficult-to-interpret results, casting shadows on its proposed health benefits [31,33,55,56,57,58,59,60]. A major limitation of previous studies is the poor understanding of resveratrol’s dose dependency, metabolism, and the mechanism underlying its effects. In addition, resveratrol has a limited bioavailability [61,62]. Furthermore, although the majority of the studies have attributed resveratrol’s benefits to its anti-oxidant effects, it also exhibits pro-oxidant effects that have been linked to its therapeutic effects [2,63,64,65,66,67,68,69,70,71,72,73,74]. Some studies suggest that whether pro-oxidant or anti-oxidant effects are exerted may depend on the concentration of resveratrol, its reactions with other compounds in the body, or the age at which the treatment is administered [72,75,76,77,78].

Studies have shown that the mixing of resveratrol and copper leads to the reduction of Cu(II) to Cu(I), generating ROS via a Fenton-like reaction in the process (Figure 1) [35,36,79,80]. ROS degrade nucleic acids in vitro and have traditionally been considered capable of damaging genomic DNA. cfChPs released from dying cells were found to induce mitochondrial damage and ROS production in living cells, suggesting that cfChPs may be a major activator of ROS [81]. Interestingly, cfChP-induced DNA damage was found to occur even in the presence of the ROS inhibitor Mito-Tempo [45], which is consistent with other recent studies that indicate that ROS may not be a damaging agent for genomic DNA [82]. Additionally, although copper-based therapeutics have been associated with toxic effects, studies by Mittra’s group have shown that a high resveratrol-to-copper ratio can achieve complete degradation of DNA; thus R-Cu combination therapy requires low concentrations of copper [83,84]. A previous study has also shown that the prolonged administration of resveratrol and copper in aged mice induces the up-regulation of anti-oxidant systems [50], which may offset any potential toxic effects arising from the generation of excess ROS. Collectively, these results suggest that the use of R-Cu to deactivate circulating cfChPs via ROS is an attractive and potentially safe therapeutic approach.

When R+Cu is taken orally, the ROS generated in the stomach is readily absorbed to have systemic effects in the form of deactivating cfChPs. The amounts of resveratrol and copper that are required to generate ROS are miniscule. In our human studies [83,85,86,87], we used 5.6 mg of resveratrol and 560 ng of copper and observed therapeutic results. On the other hand, the recommended doses are 500 mg twice a day and 2 mg once a day, for Resveratrol and copper, respectively. Therefore, our study focusses on a new chemistry wherein small amounts of Resveratrol and copper are combined to have novel therapeutic effects which are mediated via the generation of ROS.

## 4. Therapeutic Effects of R-Cu in Pre-Clinical and Clinical Studies

Various pre-clinical and clinical studies have indicated that R-Cu combination therapy can counter the damaging effects of cfChPs and thereby be an effective approach to treat multiple pathologies, including cancer.

### 4.1. Inhibitory Effect on Toxicity Related to Chemotherapy and Radiotherapy

Chemotherapy and radiotherapy remain the main therapeutic approaches utilized for cancer treatment. Both approaches are associated with toxic side-effects, including DNA damage and inflammation in healthy cells surrounding the tumor cells, which is known as the ‘bystander effect’. Using in vitro and animal experiments, Mittra’s group has shown that the numerous cfChPs released from chemotherapeutic agent-induced or radiation-induced dying cancer cells can rapidly enter into the nuclei of surrounding healthy cells and integrate into their genome and induce DNA damage and inflammation, indicating that these cfChPs promote treatment-related toxicity [41,42,48]. In these studies, the observed harmful effects could be ameliorated with concurrent use of cfChP-inactivating agents, including R-Cu [42,48]. The experimental details are summarized in Table 1.

Two clinical trials have investigated the effect of R-Cu in reducing toxicity from chemotherapy—RESCU001 [85] and RESCUIII [86]. In RESCU001 [85], a prospective single-center pilot study, 25 patients with multiple myeloma who were receiving hematopoietic stem cell transplant following high dose melphalan were given either vehicle alone (N = 5) or R-Cu at dose level I (resveratrol = 5.6 mg and copper = 560 ng; N = 15) or dose level II (resveratrol = 50 mg and copper = 5 μg; N = 5). Both doses were administered orally twice daily starting 48 h prior to melphalan chemotherapy and continued until 21 days post-transplantation. As expected, all patients in the control group developed grade 3/4 oral mucositis at the end of the treatment, whereas only 40% of patients in the R-Cu treatment groups developed mucositis. Notably, R-Cu treatment reduced the levels of inflammatory cytokines TNF-α and IL-1β at dose level I but not at dose level II. In RESCUIII [86], a single-arm phase II study, 30 patients with advanced gastric cancer who were receiving docetaxel-based multi-agent chemotherapy were administered R-Cu three times daily on an empty stomach starting one day before the start of chemotherapy and continuing for six months or until disease progression. Although the R-Cu treatment did not significantly reduce the cumulative incidence of overall and hematological toxicities, it markedly reduced the incidence of more troublesome non-hematological toxicities, including diarrhea, hand–foot syndrome, and vomiting. The methodological details are summarized in Table 1.

### 4.2. Preventive Effect on Sepsis and Viral Infections

cfChPs play a role in bacterial endotoxin-mediated sepsis [44]. Notably, R-Cu administration was reported to prevent the release of cfChPs into the extracellular spaces in an LPS-induced mouse model of sepsis, along with a reduction in the release of inflammatory cytokines, DNA damage, apoptosis, inflammation, and other hallmarks of sepsis [44]. In another preliminary observational study, R-Cu treatment reduced mortality in patients with severe COVID-19 by nearly 50% [87]. The experimental details of the above studies are summarized in Table 1. Another study reported the antiviral effects of resveratrol against SARS-CoV-2 [88]. Although this study only examined resveratrol, it is possible that the applied resveratrol may have been mobilized and combined with endogenous copper to exert pro-oxidant effects.

### 4.3. Inhibitory Effect on Aging and Degenerative Conditions

Oxidative stress is associated with aging and age-related disorders. However, this mainstream and straight-forward notion has been challenged multiple times, and anti-oxidant systems have failed to ameliorate oxidative stress-related disorders [89,90]. Furthermore, recent evidence suggests that cfChPs can continuously exert damaging effects over a lifetime, contributing to aging and age-related degeneration [43,47]. Consistent with this, Pal et al. [50] found that deactivating cfChPs using resveratrol and copper down-regulated multiple biomarkers of aging in the brain cells of C57Bl/6 mice (experimental details are presented in Table 1). These effects included a reduction in DNA damage, telomere attrition, aneuploidy, amyloid deposition, inflammation, apoptosis, senescence, and mitochondrial dysfunction, suggesting that cfChPs could be drivers of aging and age-related disorders and that R-Cu therapy could be an effective anti-aging and anti-degenerative therapeutic approach.

### 4.4. Inhibitory Effect on Metastases and Cancer Progression

Studies have also indicated that cfChPs released from dying cancer cells could be potentially oncogenic in nature and may possibly have the ability to induce metastasis [41,49]. In a recent pre-clinical study [49], NOD-SCID mice in which an MDA-MB-231 human breast cancer xenograft was generated showed the presence of human DNA signals (representing cfChPs) in their brain cells as well as the presence of eight human oncoproteins, namely c-Myc, c-Raf, p-EGFR, HRAS, p-AKT, FGFR 3, PDGFRA, and c-Abl., Localized radiotherapy, chemotherapy, and surgery markedly exacerbated the dissemination of the human DNA (cfChPs) and oncoproteins to brain cells. The concurrent administration of cfChP-deactivating agents, such as R-Cu, prevented this increased dissemination. Thus, therapeutic intervention can potentially promote metastatic spread via the systemic dissemination of oncogenes which can be prevented by concurrent treatment with R-Cu.

In an exploratory study [83], 25 patients with advanced oral squamous cell carcinoma were divided into five groups of five patients each: one control group and four R-Cu treatment groups with increasing doses of resveratrol and copper administered for two weeks. The lowest doses used were 5.6 mg and 560 ng of resveratrol and copper, and the highest doses were 500 mg and 5 mg, respectively. This study showed increased levels of cfChPs in the tumor microenvironment, which were drastically reduced at the end of the R-Cu treatment. Furthermore, the R-Cu-mediated elimination of cfChPs from the tumor microenvironment correlated with the down-regulation of ten cancer hallmarks and five immune checkpoints, with lower doses of R-Cu showing a more marked effect than higher doses and no adverse effects. Although the targeting of immune checkpoints has been a significant and successful breakthrough in cancer therapy [91], the mechanism of the up-regulation of immune checkpoints in cancer remains largely unknown. The above studies suggest that cfChPs could be instigators of immune checkpoints in cancer and that R-Cu could be an efficient means to target the activated immune checkpoints. Indeed, a recent study has shown that cfChPs released from dying cancer cells activate five immune check-points viz. NKG2A, PD-1, LAG-3, CTLA-4, and TIM-3 in human lymphocytes, an effect which was found to be abrogated in the presence of R-Cu [92].

## 5. R-Cu Induced ROS Does Not Damage Genomic DNA

Recent evidence suggests that ROS generated by mitochondria may not be a direct cause of genomic DNA damage [45]. NIH3T3 cells were treated with cfChPs which had been pre-treated with the ROS scavenger Mito-TEMPO. The result showed that while cfChPs treatment markedly increased DNA damage, this was not prevented by the presence of the ROS scavenger Mito-TEMPO. This suggested that cfChP-induced DNA damage is mediated via an ROS independent mechanism. This finding is corroborated by the results shown in Figure 2. Mice were administered R-Cu twice daily by oral gavage for two weeks at a dose of 1 mg/kg of R and 0.1 μg/kg of Cu. Thereafter, brain cells were harvested, stained with MitoSOX Red to detect mitochondrial ROS production, and subjected to fluorescent microscopy to estimate the mean fluorescence intensity (MFI) per cell. Brain sections were also stained with antibody against γH2AX to detect double-strand DNA breaks. As shown in Figure 2, R-Cu treatment resulted in a marked increase in ROS production in brain cells. However, this did not lead to an increase in DNA damage marked by γH2AX signals. On the contrary, a reduction in γH2AX signals was observed. The latter can be explained by the possibility that R-Cu treatment had deactivated the extracellular cfChPs via ROS production, thereby preventing them from causing damage to the genomic DNA of the brain cells (Figure 2).

These findings suggest that ROS generated by R-Cu can deactivate extra-nuclear DNA (i.e., cfChPs) without damaging the genomic DNA in vivo. This, along with the encouraging observations that R-Cu combination therapy has elicited no adverse effects to date [83,85,86,87], reinforces the idea that R-Cu therapy is potentially safe.

## 6. Conclusions

Substantial evidence now indicates that cfChPs released from dying cells, whether under normal or diseased conditions, have harmful effects on healthy cells in the body. Therefore, the deactivation of cfChPs promises to improve health in multiple ways. An effective way to achieve this would be to utilize the damaging effects of ROS to deactivate extracellular cfChPs and minimize their harmful effects. As the ROS itself do not damage genomic DNA, using compounds or combinations of compounds, such as resveratrol and copper, which can generate sufficient ROS to deactivate cfChPs without triggering genomic damage may be a viable and safe therapeutic approach. The studies discussed above suggest that R-Cu could be an effective adjuvant anticancer treatment to minimize the toxic effects of chemotherapy and radiotherapy and suppress the development of metastasis. In fact, considering that trace metals such as copper have been reported to be present at higher levels in cancer cells [93], the therapeutic benefits of resveratrol in cancer may be driven by its combination with copper rather than by resveratrol alone.

In conclusion, we propose that R-Cu treatment could be a potent therapeutic approach to counter multiple pathologies and retard aging and degenerative conditions.

## Figures and Tables

**Figure 1 pharmaceuticals-18-00132-f001:**
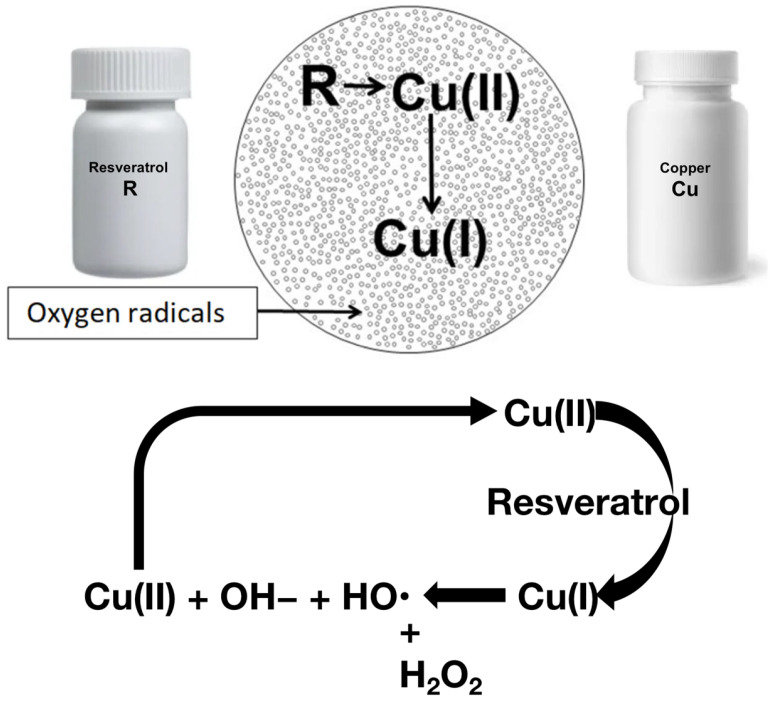
Resveratrol reacts with copper to generate oxygen radicals via a Fenton-like reaction. These oxygen radicals could underlie the pro-oxidant effects of resveratrol. Image taken from Mittra 2024 [80] under creative commons license.

**Figure 2 pharmaceuticals-18-00132-f002:**
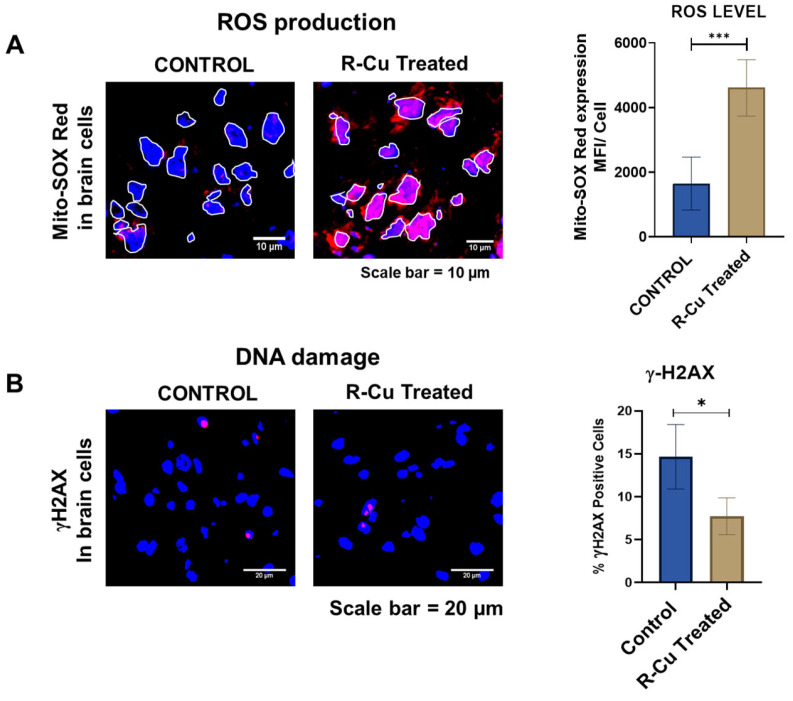
Reactive oxygen species (ROS) generated by resveratrol–copper (R-Cu) treatment does not damage genomic DNA of brain cells. Brain cells from mice administered R-Cu twice daily for two weeks at a dose of 1 mg/kg of R and 0.1 μg/kg of Cu were harvested and (**A**) stained with MitoSOX Red to detect mitochondrial ROS production or (**B**) antibody against γH2AX to detect double-strand DNA breaks. Brain sections were analyzed using fluorescent microscopy to estimate mean fluorescence intensity (MFI) per cell. For ROS estimation, the nuclei were gated to exclude the interfering nuclear fluorescence from the MFI analysis. Asterisks indicate significant difference compared to the control. *** *p* < 0.001 * *p* < 0.05. This experiment was performed twice.

**Table 1 pharmaceuticals-18-00132-t001:** Summary of studies showing the efficacy of resveratrol and copper (R-Cu) combination therapy to overcome the damaging effects of cell-free chromatin particles (cfChPs) in various pathological conditions.

Type of Study	Condition Examined	Model/Patients	Treatment/Intervention	Results	Reference
Pre-clinical	Chemotherapy-induced toxicity	C57BL/6 female mice	Mice were treated with control (saline i.p. b.d); single sub-lethal dose of adriamycin (10 mg/ kg, i.p.); or R-Cu (R 1 mg/kg and Cu 10^−4^ mg/kg by oral gavage, b.d.) + adriamycin (4 h after R-Cu).	R-Cu treatment inhibited chemotherapy (adriamycin)-induced tissue DNA damage, apoptosis, and inflammation in multiple organs and peripheral blood mononuclear cells. It prevented prolonged neutropenia following a single adriamycin dose and reduced the death rate post lethal dose of adriamycin.	[42]
Pre-clinical	Radiation-induced toxicity	BALB/c mice	Mice were subjected to lower hemi-body irradiation (HBI; 10 Gy) with or without R-Cu (R = 1 mg/kg and Cu = 10^−4^ mg/kg twice daily by oral gavage; the first dose of R-Cu was given 4 h prior to HBI)	Radiation-induced activation of bystander effect biomarkers (H2AX, active Caspase-3, NFκB, and IL-6) in the brain cells was prevented by co-treatment of R-Cu.	[48]
Pre-clinical	Sepsis	C57BL/6 female mice	Mice were administered a single i.p. injection of LPS at a dose of 10 mg/kg or 20 mg/kg with or without concurrent treatment with R-Cu (R = 1 mg/kg and Cu = 10^−4^ mg/kg; administered 4 h prior to LPS challenge)	R-Cu treatment abrogated the following effects of LPS (i) release of cfCh in extra-cellular spaces of brain, lung, and heart and in circulation; (ii) release of inflammatory cytokines; (iii) activation of DNA damage, apoptosis and inflammation in cells of thymus, spleen and in PBMCs; (iv) DNA damage, apoptosis, and inflammation in cells of lung, liver, heart, brain, kidney, and small intestine; (v) liver and kidney dysfunction and elevation of serum lactate; (vi) coagulopathy, fibrinolysis, and thrombocytopenia; (vii) lethality.	[44]
Clinical	COVID-19	Patients with severe COVID-19 requiring inhaled oxygen	Of 230 patients, 30 received R and Cu in addition to standard care at doses of 5.6 mg and 560 ng, respectively, orally, once every 6 h, until discharge or death.	Binary logistic regression analysis revealed a trend towards a reduction (nearly two-fold) in death in patients receiving R-Cu.	[87]
Clinical	Advanced squamous cell carcinoma of oral cavity	Patients with advanced oral cancer	Of 25 patients, 5 acted as controls and the remaining 20 were given R-Cu in increasing doses, with the lowest dose of R-Cu being 5.6 mg and 560 ng, respectively, and the highest dose being 500 mg and 5 mg, respectively. An initial biopsy was taken from patients at first presentation, and a second biopsy was taken 2 weeks later. R-Cu was administered orally twice daily in the intervening period.	R-Cu treatment reduced cfChPs in the tumor microenvironment and down-regulated 21/23 biomarkers of cancer, with no adverse effects observed. The lower two doses of R-Cu were more effective than the higher doses.	[83]
Pre-clinical	Aging and neurodegenration	C57BL/6 mice	Of 24 mice, 4 were sacrificed when they were 3 months old (young controls). Of the remaining 16, at 10 months old, 8 were treated with R-Cu (R = 1 mg/kg and Cu = 10^−4^ mg/kg) twice daily by oral gavage for 12 months and 8 acted as controls. All 16 were sacrificed after 12 months at 22 months- old.	R-Cu treatment reduced the hallmarks of aging, including telomere attrition, amyloid deposition, DNA damage, apoptosis, inflammation, senescence, aneuploidy, and mitochondrial dysfunction	[50]
Clinical	Bone marrow transplant-related toxicity	Patients with multiple myeloma receiving hematopoietic stem cell transplant with high dose melphalan	Of 25 patients, 5 acted as controls; the remaining 20 received R-Cu twice daily, at dose level I (DL-I; R = 5.6 mg and Cu = 560 ng; N = 15); and DL-II (R = 50 mg and Cu = 5 μg; N = 5).	R-Cu treatment reduced transplant-related toxicities (incidence of grade 3/4 oral mucositis, levels of inflammatory cytokines)	[85]
Clinical	Chemotherapy-related toxicity in gastric cancer	Patients with advanced gastric cancer receiving docetaxel-based multi-agent chemotherapy	Patients were treated with one of two chemotherapeutic regimens: (1) TEX every 2 weeks [docetaxel (50 mg m^−2^ iv on day 1), oxaliplatin (85 mg m^−2^ iv on day 1) and capecitabine (1000 mg m^−2^, P.O, on days 1–14)] and (2) DOF every 2 weeks [docetaxel (50 mg m^−2^ iv on day 1, oxaliplatin (85 mg m^−2^ iv on day 1) and 5-fluoro-uracil (1200 mg m^−2^, i.v., on days 1–2 via infusional pump) + leucovorin (200 mg m^−2^ i.v. day 1)]. Patients were administered R-Cu thrice daily 1 h before meals starting one day before the start of chemotherapy. R-Cu was administered for 6 months or till first evidence of disease progression.	R-Cu treatment reduced the incidence of non-hematological toxicities (hand–foot syndrome, diarrhea, and vomiting) without adversely affecting progression-free and overall survival rates. Note that R-Cu treatment did not reduce the overall cumulative incidence of grade ≥ 3 toxicity or of ≥ 3 hematological toxicity.	[86]

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
