# Peer review of "Copper Imparts a New Therapeutic Property to Resveratrol by Generating ROS to Deactivate Cell-Free Chromatin"

_pharmaceuticals, 2025, doi:10.3390/ph18010132_

Round 1

Reviewer 1 Report

Comments and Suggestions for Authors

Comments and Suggestions

The Perspective article by Mittra and Khanvilkar highlights the potential of resveratrol and copper as a combination therapy for various diseases through the deactivation of cell-free chromatin particles (CfChPs). The authors primarily focus on the role of reactive oxygen species (ROS) generated by the resveratrol-copper complex as key therapeutic agents.

The article provides a comprehensive and well-structured discussion of CfChPs, resveratrol, and the resveratrol-copper complex, clearly illustrating their implications in human health care. The authors have done an excellent job of integrating relevant information, offering valuable insights into this emerging therapeutic approach. I believe this article will attract a broad readership and would be a valuable addition to Pharmaceutical, so I recommend its publication.

I have a few minor concerns and suggestion, which are as follows: 

1.         Please consider summarizing the implications of resveratrol, copper, and resveratrol-copper complexes in human healthcare within a single paragraph. Additionally, provide an overview of their applications, supported by statistical data trends over the year.

2.         Please include a schematic representation illustrating how CfChPs generated from dying cells enter healthy cells. The diagram should highlight their interference with major biochemical pathways, ultimately leading to various disease conditions. This visual aid would enhance the reader’s understanding of the underlying mechanisms.

3.         It is unclear whether the deactivation of CfChPs is exclusively mediated by ROS generated specifically by the resveratrol-copper complex, or if it can be achieved by ROS from any source. If the latter is the case, it could potentially diminish the significance of the resveratrol-copper complex in this context. Please clarify this point.

4.         Page 2, lines 41–62: The statement, “Thus, CfChPs may potentially contribute to metastases as well as the harmful bystander effects observed during cancer treatment,” is vague as it lacks specific details. Metastasis is a complex process involving numerous biomolecules and signaling pathways. Consider elaborating on the specific mechanisms or pathways through which CfChPs might contribute to these processes to provide greater clarity and depth.

5.         It is unclear whether resveratrol binds to specific receptors or other biomolecules within cells. If there are any relevant literature reports on such interactions, please highlight them. Otherwise, clarify whether its effects are solely attributed to its pro- or anti-oxidant properties.

6.         Page 4, lines 108–113: The statements in this section are somewhat confusing. It is mentioned that CfChPs trigger ROS production in healthy cells, followed immediately by a statement about deactivating CfChPs via RIO. Please elaborate and clarify how these processes are connected to provide a more coherent explanation.

7.         Consider providing a comprehensive table for Section 4 that highlights the role of the resveratrol-copper complex (R-Cu) in various disease conditions. This would improve clarity and allow readers to easily understand the therapeutic implications across different contexts.

Author Response

Reviewer 1

The Perspective article by Mittra and Khanvilkar highlights the potential of resveratrol and copper as a combination therapy for various diseases through the deactivation of cell-free chromatin particles (CfChPs). The authors primarily focus on the role of reactive oxygen species (ROS) generated by the resveratrol-copper complex as key therapeutic agents.

The article provides a comprehensive and well-structured discussion of CfChPs, resveratrol, and the resveratrol-copper complex, clearly illustrating their implications in human health care. The authors have done an excellent job of integrating relevant information, offering valuable insights into this emerging therapeutic approach. I believe this article will attract a broad readership and would be a valuable addition to Pharmaceutical, so I recommend its publication.

Response: Thank you for your positive remarks.

I have a few minor concerns and suggestion, which are as follows: 

  1. Please consider summarizing the implications of resveratrol, copper, and resveratrol-copper complexes in human healthcare within a single paragraph. Additionally, provide an overview of their applications, supported by statistical data trends over the year.

Response: Individually, both resveratrol and copper have been extensively studied for their therapeutic effects, and we have also extensively referred to the relevant studies and reviews in the Introduction section and later on (Refs #3, 6–30, 51, 52). However, our current article focuses specifically on the combination of R+Cu and its therapeutic effects by deactivating cfChPs via the medium of ROS. For the therapeutic effects of R-Cu combination, we have included detailed descriptions of pre-clinical and clinical studies under Section 4 of the manuscript. In addition, we have also summarized these results in a newly added Table (Table 1).

  1. Please include a schematic representation illustrating how CfChPs generated from dying cells enter healthy cells. The diagram should highlight their interference with major biochemical pathways, ultimately leading to various disease conditions. This visual aid would enhance the reader’s understanding of the underlying mechanisms.

Response: Thank you for this suggestion. We have prepared an animated video (Video S1) to illustrate the entry of CfChPs into healthy cells and its consequences, mainly DNA damage and activation of inflammation and apoptosis processes, and how these lead to multiple disease conditions. We hope the animation will enhance the understanding of the effects of cfChPs.

  1. It is unclear whether the deactivation of CfChPs is exclusively mediated by ROS generated specifically by the resveratrol-copper complex, or if it can be achieved by ROS from any source. If the latter is the case, it could potentially diminish the significance of the resveratrol-copper complex in this context. Please clarify this point.

Response: It is possible that ROS is also generated from other sources, but their therapeutic effects have neither been investigated nor reported. Since our studies indicated that ROS generated from the chemistry between R and Cu can efficiently deactivate cfChPs, we have focused specifically on the therapeutic effects of ROS generated by R + Cu.

         Page 2, lines 41–62: The statement, “Thus, CfChPs may potentially contribute to metastases as well as the harmful bystander effects observed during cancer treatment,” is vague as it lacks specific details. Metastasis is a complex process involving numerous biomolecules and signaling pathways. Consider elaborating on the specific mechanisms or pathways through which CfChPs might contribute to these processes to provide greater clarity and depth.

Response: We agree that metastasis is thought to be a complex process. However, the mechanisms underlying the various steps of the metastatic cascade have not been fully elucidated. Herein, we are proposing an alternative model of cancer metastasis whereby cfChPs released from dying cancer cells enter the circulation and are taken up by cells of distant organs which are transformed to form new tumours which masquerade as metastasis. We have clarified and discussed this issue in detail in section 4.4. (pp. 5, lines 200 – 209).

  1. It is unclear whether resveratrol binds to specific receptors or other biomolecules within cells. If there are any relevant literature reports on such interactions, please highlight them. Otherwise, clarify whether its effects are solely attributed to its pro- or anti-oxidant properties.

Response: As mentioned above we are not focusing on the effects of resveratrol on its own, and therefore not discussed its binding to receptor or other intracellular biomolecules. Rather, we are focusing on the chemistry between R and Cu, which when combined together generate ROS. The latter then have systemic therapeutic effects by deactivating cfChPs.

  1. Page 4, lines 108–113: The statements in this section are somewhat confusing. It is mentioned that CfChPs trigger ROS production in healthy cells, followed immediately by a statement about deactivating CfChPs via RIO. Please elaborate and clarify how these processes are connected to provide a more coherent explanation.

Response: We thank the reviewer for pointing out this confusion. We have now revised the entire paragraph to clarify the issues involved (Page 3, lines 107 – 136).   

  1. Consider providing a comprehensive table for Section 4 that highlights the role of the resveratrol-copper complex (R-Cu) in various disease conditions. This would improve clarity and allow readers to easily understand the therapeutic implications across different contexts.

Response: We thank the reviewer for this suggestion. We have now prepared a table summarizing all the pre-clinical and clinical studies with R-Cu including the experimental details (Table 1).

We would like to point out that we have modified the section heading 2, 3 and 4 which now read as follows.

Section 2: The damaging effects of cfChPs (this section has been completely revised)

Section 3: Resveratrol-Copper (R-Cu) Combination as an Effective Therapeutic Solution to Deactivate cfChPs

Section 4: Therapeutic effects of R-Cu in pre-clinical and clinical studies.

Reviewer 2 Report

Comments and Suggestions for Authors

Comments on the manuscript pharmaceuticals-3303777 by Salooni Khanvilkar and Indraneel Mittra titled “Copper imparts a new therapeutic property to resveratrol by generating ROS to deactivate cell-free chromatin”:

The presented perspective article provides some information on the possible positive effects of combination resveratrol-copper in the therapy of cell-free chromatin-linked diseases (cardiovascular disease, diabetes, Alzheimer's disease, multiple types of cancers, etc.). The first question is why copper? Is there evidence of a similar effect caused by other metal? Furthermore, it is unclear whether the idea to additionally generate ROS contradicts the concept that the balance between ROS and antioxidants is optimal, as both extremes, oxidative and antioxidative stress, are damaging. Resveratrol itself possesses diverse biological properties, including neuroprotective, antitumor, anti-inflammatory, and osteoporosis inhibition effects (10.3389/fphar.2024.1417532). The molecular basis of the action of resveratrol is extremely complex, and considering only the antioxidant properties of this compound is very speculative. The authors do not propose any mechanism of action - the reaction presented in Figure 1 is not informative at all. It is not clear how the problem of low bioavailability of resveratrol will be overcome, nor are its adverse effects addressed.

Author Response

Reviewer 2

The presented perspective article provides some information on the possible positive effects of combination resveratrol-copper in the therapy of cell-free chromatin-linked diseases (cardiovascular disease, diabetes, Alzheimer's disease, multiple types of cancers, etc.). The first question is why copper? Is there evidence of a similar effect caused by other metal? 

Response: We took the cue from the seminal studies by Fukuhara et. al. that Resveratrol (R) when combined with copper (Cu) generates ROS, which can cleave plasmid pBR322 DNA (Refs. 35 and 36 in the manuscript). By extending their findings, we have discovered that R + Cu can deactivate the DNA component of the cell-free chromatin particles (cfChPs) via ROS and that cfChPs deactivation has multiple therapeutic effects (Refs 42, 44, 48, 50, 83, 85, 86, 87 and Table 1). It is possible that resveratrol in combination with other metals may also generate ROS; but we have not pursued other combinations since we found R-Cu to be highly therapeutically effective and potentially safe.

Furthermore, it is unclear whether the idea to additionally generate ROS contradicts the concept that the balance between ROS and antioxidants is optimal, as both extremes, oxidative and antioxidative stress, are damaging.

Response: The fact that ROS generated by R+Cu does not damage genomic DNA or cause any adverse effects (in animal and human studies) suggests that it might not generate sufficient ROS to create oxidative stress.

Resveratrol itself possesses diverse biological properties, including neuroprotective, antitumor, anti-inflammatory, and osteoporosis inhibition effects (10.3389/fphar.2024.1417532). The molecular basis of the action of resveratrol is extremely complex, and considering only the antioxidant properties of this compound is very speculative.

Response: We agree that Resveratrol has been claimed to have multiple therapeutic effects which we have extensively quoted (Refs 3, 6–30 in the manuscript). However, many questions remain viz. poor bioavailability, limited understanding of its metabolism and dose-dependency, potential adverse effects and, above all, highly inconsistent results across pre-clinical and clinical studies [31–34]. Furthermore, we are not discussing resveratrol’s potential antioxidant therapeutic properties; rather we are claiming that resveratrol acts as a pro-oxidant when combined with copper and that this combination has multiple therapeutic effects that result from ROS-mediated deactivation of cfChPs. We have now summarized the therapeutic effects of R-Cu in section 4 and the accompanying Table 1

The authors do not propose any mechanism of action - the reaction presented in Figure 1 is not informative at all. 

Response: The purpose of presenting Figure 1 is to illustrate how R + Cu generates ROS. The mechanism of action is now represented in a video format (Video S1).

It is not clear how the problem of low bioavailability of resveratrol will be overcome, nor are its adverse effects addressed.

Response: The focus of this article is not Resveratrol, rather it is the role of Resveratrol + copper which leads to the generation of ROS to deactivate cfChPs with multiple therapeutic effects. Therefore, the question of poor bioavailability of resveratrol is not relevant in our context.

When R+Cu is taken orally, the ROS generated in the stomach is readily absorbed to have systemic effects in the form of deactivating cfChPs. The amounts of resveratrol and copper that are required to generate ROS are miniscule. In our human studies (Refs 83, 85, 86, 87), we used 5.6 mg of resveratrol and 560 ng of copper and observed therapeutic results. On the other hand, the recommended doses are 500 mg twice a day and 2 mg once a day, for Resveratrol and copper respectively. Therefore, our study focusses on a new chemistry when small amounts of Resveratrol and copper are combined to have novel therapeutic effects mediated via the generation of ROS.

We have now incorporated the above issue in the manuscript (pp. 4, lines. 129 - 136)

We would like to point out that we have modified the section heading 2, 3 and 4 which now read as follows.

Section 2: The damaging effects of cfChPs (this section has been completely revised)

Section 3: Resveratrol-Copper (R-Cu) Combination as an Effective Therapeutic Solution to Deactivate cfChPs

Section 4: Therapeutic effects of R-Cu in pre-clinical and clinical studies.

Round 2

Reviewer 2 Report

Comments and Suggestions for Authors

The revised version of the manuscript can be accepted without changes.

Author Response

The revised version of the manuscript can be accepted without changes.